# Vitamins K2 and D3 Improve Long COVID, Fungal Translocation, and Inflammation: Randomized Controlled Trial

**DOI:** 10.3390/nu17020304

**Published:** 2025-01-16

**Authors:** Ornina Atieh, Joviane Daher, Jared C. Durieux, Marc Abboud, Danielle Labbato, Jhony Baissary, Ziad Koberssy, Kate Ailstock, Morgan Cummings, Nicholas T. Funderburg, Grace A. McComsey

**Affiliations:** 1School of Medicine, Case Western Reserve University, Cleveland, OH 44106, USA; ornina.atieh@case.edu (O.A.); joviane.daher@case.edu (J.D.); jxb1120@case.edu (J.B.); ziad.koberssy@case.edu (Z.K.); 2Clinical Research Center, University Hospitals Cleveland Medical Center, Cleveland, OH 44106, USA; jared.durieux@uhhospitals.org (J.C.D.); danielle.labbato@uhhospitals.org (D.L.); 3Faculty of Medicine, Saint Joseph University, Beirut, Lebanon; marc.abboud@net.usj.edu.lb; 4School of Health and Rehabilitation Sciences, Ohio State University, Columbus, OH 43210, USA; kate.ailstock@osumc.edu (K.A.); morgan.cummings@osumc.edu (M.C.); nicholas.funderburg@osumc.edu (N.T.F.)

**Keywords:** long COVID, vitamin K2/D3 supplementation, gut permeability, fungal translocation, chronic inflammation

## Abstract

Background: Long COVID (LC) is characterized by persistent symptoms at least 3 months after a SARS-COV-2 infection. LC has been associated with fungal translocation, gut dysfunction, and enhanced systemic inflammation. Currently, there is no approved treatment for this condition. The anti-inflammatory effect of vitamins K2 and D3 was shown to help attenuate the course of acute COVID-19 infection. Objective and hypothesis: This trial aims to investigate the effects of vitamins K2/D3 on LC symptoms, as well as gut and inflammatory markers, in people with established long COVID. Our hypothesis is that by attenuating systemic inflammation, vitamins K2/D3 will improve long COVID symptoms. Methods: This single-site randomized controlled study enrolled adults experiencing ≥2 moderate LC symptoms at least 3 months after a COVID-19 infection. The RECOVER Long COVID Research Index and number and type of LC symptoms were considered. Participants were randomized 2:1 to daily 240 µg K2 (pure MK-7 form) and 2000 UI vitamin D3 or standard of care (SOC) for 24 weeks. The endpoints were changes in symptomatology and in select inflammatory, metabolic, and gut biomarkers at 24 weeks. Results: We enrolled 151 participants (*n* = 98 received vit K2/D3 and 53 received SOC). The median age was 46 years; 71% were female and 29% were non-white. Baseline demographics were balanced between groups. At 24 weeks, the active treatment group only had a sharp increase in 25(OH) D, indicating good treatment adherence. In the vitamin K2/D3 arm, there was a 7.1% decrease in the proportion who had an LC Index ≥12 (vs. a 7.2% increase in the SOC group; *p* = 0.01). The average number of LC symptoms remained stable in the vitamin K2/D3 arm but increased in the SOC arm (*p* = 0.03). Additionally, reductions in oxidized LDL, inflammatory markers sTNF-RI and sCD163, and fungal translocation marker (1,3)-β-d-glucan were observed in the vitamin K2/D3 arm compared to the SOC arm (*p* < 0.01) over 24 weeks. Conclusions: Vitamins K2/D3 improved the RECOVER Long COVID Index, the number of LC symptoms, and several gut and inflammatory markers. Vitamins K2/D3 provide a promising safe intervention for people suffering from long COVID.

## 1. Introduction

With millions of confirmed cases and attributable deaths, the coronavirus disease 2019 (COVID-19), caused by the severe acute respiratory syndrome coronavirus (SARS-CoV-2), remains a critical worldwide concern for healthcare professionals [1]. In addition to its initial presentation, the infection can be complicated by long COVID (LC). LC is an emerging clinical entity characterized by persistent, relapsing, or new symptoms occurring at least 3 months after acute infection [2,3].

Long COVID can present with a multitude of symptoms, most commonly including post-exertional malaise, fatigue, brain fog, dizziness, gastrointestinal issues, and palpitations [3]. With a prevalence of no less than 6% [4], LC represents a significant detriment to the quality of life for many patients [5], thus highlighting the importance of effective management strategies. This need is further emphasized by the current absence of approved treatments—whether antiviral or anti-inflammatory—for this condition.

Currently, the pathophysiology behind LC remains under investigation, with some studies implicating viral persistence and ongoing inflammatory state with a significant elevation of peripheral and central cytokines as possible mechanisms [6,7,8]. Additionally, oxidized low-density lipoprotein (ox-LDL), a major driver of atherosclerosis, has been linked to LC [9], potentially further increasing the endothelial dysfunction and inflammatory response via monocyte/macrophage-trained immunity [10]. Furthermore, systemic inflammation is exacerbated by microbial translocation due to the disruption of the gut–lung axis during acute COVID-19 and other respiratory diseases [11,12]. This has prompted further research, revealing a significant increase in the fungal cell wall polysaccharide (1,3)-β-d-glucan (BDG) and in the intestinal permeability marker Zonulin in people with LC. This increase indicates that alteration in gut integrity and fungal translocation may be associated with LC [13,14].

Given its burden, the COVID-19 pandemic has prompted extensive research into various therapeutic approaches to help regulate the severity of the disease. Among the promising interventions are the roles of vitamins K2 and D3, with their excellent safety profile and anti-inflammatory properties [15,16].

Vitamin D plays a crucial role in regulating immune cells, including dendritic cells, monocytes/macrophages, T cells, and B cells, influencing both the inflammatory response and infection protection [15,17,18]. It enhances the differentiation of monocytes and macrophages, leading to the production of antimicrobial peptides. It also modifies T cell responses by promoting anti-inflammatory cytokines while suppressing pro-inflammatory pathways [17,18]. Specifically, vitamin D influences T cell maturation away from the inflammatory Th17 phenotype and diminishes the production of various inflammatory cytokines including TNFα, while increasing anti-inflammatory cytokines such as IL-10 [19,20]. In addition, vitamin K can modulate the inflammatory response by reducing IL-6 and protecting vascular and pulmonary tissues from inflammation-related damage during SARS-CoV-2 infection [21,22].

In previous studies, vitamin D and K deficiencies have been associated with severe COVID-19 outcomes [23,24,25,26], while vitamin D-sufficient patients showed a better clinical outcome [27]. Moreover, a recent meta-analysis showed that supplementation with vitamin D3 may potentially reduce the risk of ICU admission and death associated with acute COVID-19 [28].

Long COVID is associated with an ongoing inflammatory state and the absence of effective treatments. Vitamins K2 and D3 have demonstrated anti-inflammatory effects and a favorable safety profile. Thus, we aim to investigate the impact of the supplementation of vitamins K2 and D3 on long COVID symptoms, and on biomarkers of systemic inflammation, gut integrity, and fungal translocation.

## 2. Materials and Methods

### 2.1. Study Design

This study is a single-site randomized controlled clinical trial conducted at University Hospitals Cleveland Medical Center (UHCMC) in Cleveland, OH, USA, from 2022 to 2024. Participants with long COVID were randomly assigned 2:1 to receive either 24 weeks of co-formulated vitamins K2/D3 or standard of care. The study protocol received approval from the UHCMC IRB (STUDY20220109; approval date: 8 March 2022). Prior to enrollment, each participant provided signed informed consent. The study drug was provided to participants free of charge and was donated in bulk at the start of the study by CanPrev Natural Health Products, Ltd., Scarborough, ON, Canada. Vitamin K2 (MK-7) was provided by Kappa Bioscience AS, Oslo, Norway.

### 2.2. Study Population

Participants were 18 years or older and reported at least two moderate-intensity (affecting daily life) symptoms attributable to long COVID. Symptoms must be new since the acute COVID-19 episode and must remain persistent for at least three months following the confirmed COVID-19 infection. Confirmation was obtained through documented positive Nucleic Acid Amplification Test (NAAT) PCR or an authorized SARS-CoV-2 antigen test. Additionally, individuals who tested positive for nucleocapsid or spike antibodies (for those unimmunized) were eligible, provided their symptoms were persistent for at least three months after antibody detection. For those who reported taking any vitamin D supplements at the time of screening, eligibility required documentation of 25-hydroxyvitamin-D (25(OH)D) levels of 30 ng/mL or lower [29] within the past 30 days and willingness to discontinue their vitamin D during the study. Furthermore, women of childbearing potential had to provide a negative serum or urine pregnancy test within 72 h prior to entry.

The exclusion criteria included pregnant or lactating women, individuals with known allergies or intolerances to vitamin D3 or K2, those undergoing treatment with vitamin K antagonists, and participants with a BMI of less than 18 kg/m^2^. Additionally, individuals with active neoplastic diseases requiring chemotherapy or the use of immunosuppressive medications, those who have been hospitalized in the past 28 days, and participants taking agents that might affect inflammation in the preceding three months were excluded.

### 2.3. Sample Size

In order to detect a 20% change in makers of inflammation between treatment groups with 2:1 allocation, a type I error of 0.05, and a 0.80 probability of type II error, we estimated a total sample size of 146 [D3/K2 arm (*n* = 100); SOC arm (*n* = 46)].

### 2.4. Randomization and Masking

The study statistician implemented a 2:1 randomization scheme using a simple randomization method. The study team was blinded to the randomization schedule until the time of treatment assignment. An investigational drug pharmacy reviewed and approved the study dispensing.

### 2.5. Intervention Details

Participants underwent a screening visit to confirm eligibility before randomization. Those who successfully passed screening were randomly allocated to receive the study drug or standard of care. Subjects in the active group received two daily capsules of co-formulated vitamin D3 and K2 (MK-7 form), providing a total daily dose of 2000 IU of vitamin D3 and 240 µg of vitamin K2-MK7 (K2VITAL^®^MCT) (each capsule containing 1000 IU D3 and 120 µg K2) for 24 weeks. Participants randomized to the control or standard of care group were continued to be managed by their clinical provider outside of the study and did not receive any study drug or placebo as part of their participation in the study.

### 2.6. Study Procedures

At each study visit (weeks 0, 12, and 24), participants underwent a focused physical examination that included measurements of height, weight, and vital signs. Medical and family histories were obtained from both individuals and their electronic health records. Additional questionnaires were administered to gather details on physical activity, substance use, and the intake or changes in medications and over-the-counter supplements.

Blood samples obtained after a fast of at least 12 h, as well as detailed long COVID symptom questionnaires, were also collected at baseline (week 0), 12, and 24 weeks. LC symptoms were categorized into the following three groups: general symptoms (such as fatigue, fever, chills, and gastrointestinal symptoms), neurocognitive symptoms (e.g., brain fog and headaches), and cardiopulmonary symptoms (such as breathing difficulties, coughing, and palpitations).

### 2.7. Study Measurements and Outcomes

At baseline, trained healthcare professionals used a standardized questionnaire to gather information on participants’ smoking status and demographic characteristics, including age, race/ethnicity, and sex at birth. The RECOVER Long COVID Research Index was calculated at both baseline and the end of the intervention, using an optimal threshold of 12 or higher [3]. Additionally, the number and type of LC symptoms was recorded at baseline and was reassessed at the end of the study.

Adherence to the study intervention was evaluated at each visit through comprehensive pill counts

### 2.8. Vitamin D Status and Metabolic Biomarkers

Fasting blood samples were used at each visit to assess the status of vitamin D using a measurement of 25-hydroxyvitamin D levels (25(OH)D), fasting glucose, insulin—with the derived estimate of insulin resistance HOMA-IR—and lipids (measured at a CLIA-certified laboratory at University Hospitals Cleveland Medical Center, Cleveland, OH, USA).

### 2.9. Inflammatory and Gut Integrity Biomarkers

Blood samples were processed within two hours, aliquoted, and stored at −80 °C until they were shipped on dry ice to Dr. Funderburg’s laboratory at Ohio State University for biomarker measurement using an enzyme-linked immunosorbent assay (ELISA). The following biomarkers were measured: soluble CD14 and CD163 (sCD14 and sCD163) as markers of monocyte activation, high-sensitivity C-reactive protein (hs-CRP), interleukin-6 (IL-6), inducible protein of 10 kDa (IP-10), tumor necrosis factor receptors 1 and 2 (TNF-RI and TNF-RII), and intercellular adhesion molecule-1 (ICAM). Measurements were conducted using kits from R&D Systems (Minneapolis, MN, USA). Additionally, oxidized LDL (oxLDL) was measured using kits from Mercodia (Uppsala, Sweden) and D-dimer was measured using kits from Diagnostica Stago (Parsippany, NJ, USA). We also measured Zonulin (Promocell, Heidelberg, Germany) as a marker of gut permeability, Intestinal Fatty Acid Binding Protein (IFABP) (R&D Systems) as marker of intestinal integrity, and lipopolysaccharide-binding protein (LBP) (R&D Systems) and (1,3)-β-d-glucan (BDG) as markers of bacterial translocation and fungal translocation, respectively.

### 2.10. Statistical Analysis

All participants who were randomized and had a baseline visit were considered in this analysis, regardless of protocol compliance. Continuous variables were described using mean ± standard deviation or median and interquartile range (IQR), and categorical variables were described using number (*n*) and percentage (%). Differences in baseline characteristics between groups were computed using an independent *t*-test, chi-square, or Fisher’s exact test. A constrained longitudinal analysis of covariance models was used to assess the treatment effect of active vitamins K2/D3 compared to the standard of care over six months, with a first-order autoregressive correlation structure. Generalized linear mixed models with a random intercept, the logit link function, and a binary distribution were used to estimate time-varying models for binary outcome, Long COVID status was assessed using the marginal maximum likelihood test with adaptive Gaussian quadrature. Adjusted models included treatment arm, age, serum vitamin D at baseline, sex, race, BMI, lipids, D-dimer, and BDG. Changes in markers of inflammation (ICAM, TNF-RI, oxLDL, and sCD163) were modeled separately from each other. Log transformations were used to reduce error variance, and all analyses were performed using SAS 9.4 (SAS Inc., Cary, NC, USA). *p*-values less than alpha < 0.05 were considered statistically significant.

## 3. Results

### 3.1. Study Sample at Baseline

A total of 247 individuals were screened for study eligibility. Among them, 85 did not meet inclusion criteria, and 11 declined to participate. The intent-to-treat population included 151 volunteers with long COVID who were randomized to either the vitamin K2/D3 arm (*n* = 98) or the standard of care arm (SOC) (*n* = 53) (Figure 1). Among the vitamin K2/D3 arm, the average age was 45 ± 12.9 years, 70.4% were female sex, and 31% were non-white race (Table 1). At baseline, the distribution of age, sex, race, BMI, glucose, and lipids was similar (*p* > 0.05) between the study arms. None of the study participants reported any chronic inflammatory gut conditions.

At the time of treatment allocation, 19.2% of the vitamin K2/D3 arm and 8.6% of the SOC arm had serum 25(OH)D levels of <20 ng/mL (*p* = 0.8), and 42.9% of the vitamin K2/D3 arm (vs. SOC: 54.7%; *p* = 0.2) had an LC Research Index of ≥12 (Appendix A). The most frequently reported symptoms were fatigue [K2/D3 (82.7%); SOC (97.6%)], problems thinking [K2/D3 (74.1%); SOC (92.9%)], pain in any part of the body [K2/D3 (61.7%); SOC (40.5%)], shortness of breath [K2/D3 (55.6%); SOC (69%)], and change in smell or taste [K2/D3 (45.7%); SOC (64.3%)]. Overall, the average number of reported LC symptoms [K2/D3 = 8.1 ± 5.5 vs. SOC = 9.5 ± 6.1] and the proportion of participants who experienced symptoms characteristic of general, neurocognitive, and cardiopulmonary phenotypes were similar (*p* > 0.05) between the treatment arms.

### 3.2. Adverse Events

Over the study period, 91 participants (*n* = 62 in the vitamin arm and 29 in the SOC arm) reported a total of 213 adverse events (AEs) (Appendix A). Overall, 28 participants prematurely discontinued the study, with two of these being related to adverse events (headache Grade 1 and myocardial infarction). AEs were classified by severity as follows: Grade 1 (*n* = 159), Grade 2 (*n* = 49), Grade 3 (*n* = 4), and Grade 4 (*n* = 1). No AEs were attributed to the study intervention.

### 3.3. Adherence to Study Intervention

A pill count was obtained at each study visit. Adherence was calculated, and the median was 91% (range 26–100) between entry and the visit at 12 weeks, and 86% (range 13–100) for the period between the 12- and 24-week visits.

### 3.4. Long COVID Symptoms

At the 6-month mark of the study, in the vitamin K2/D3 arm, there was a 7.1% decrease in the proportion who had an LC Research Index ≥ 12 and a 4.9% decrease in the proportion who had ≥2 symptoms. Additionally, there was a 15–20% reduction in reporting change in smell or taste, pain in any part of the body, persistent cough, post-exertional malaise, and shortness of breath. In contrast, in the SOC arm, there was a 7.2% increase in the proportion who had an LC Research Index ≥ 12 and no change in the proportion who reported ≥2 symptoms. At month six, the average number of reported LC symptoms remained stable in the vitamin K2/D3 arm (8.8 ± 5.6) but increased (10.9 ± 5.8) in the SOC arm (Figure 2).

### 3.5. Changes in Metabolic Biomarkers

At baseline, there were no significant differences between the vitamin K2/D3 and SOC groups in BMI, fasting lipids, glucose, or HOMA-IR. Over 24 weeks, these parameters remained stable in both groups, with no significant changes observed between treatment arms. At 24 weeks, the average fasting glucose level for the vit DK2/D3 group increased by 5.4 ± 21.1 mg/dL vs. a decrease of 6.3 ± 26.5 mg/dL (*p* = 0.001).

In univariate models, the average serum 25(OH)D increase over six-months was 10.7 mg/dL and the proportion with serum 25(H)D < 20 ng/mL levels decreased 12.6% in the vitamin K2/D3 arm. Among the SOC arm, the proportion with serum 25(OH)D levels < 20 ng/mL increased by 3.8% at 24 weeks. Overall, the estimated rate of change in 25(OH)D over time (Figure 3) was greater among the vitamin K2/D3 arm compared to the SOC arm (*p* < 0.0001).

### 3.6. Changes in Biomarkers and Associations with Changes in LC

Reductions in oxidized LDL, sTNF-RI, sCD163, and the fungal translocation marker (1,3)-β-d-glucan (BDG) were observed in the vitamin K2/D3 arm compared to the SOC arm (*p* < 0.01) over 24 weeks (Figure 4). In the vitamin K2/D3 arm, the mean oxLDL decreased by 9148.0 ± 28,445.7 mU/L [vs. SOC (−6633.4 ± 3400.9); *p* = 0.001], sTNF-RI decreased by 106.3 ± 239.0 pg/mL [vs. SOC (−91.9 ± 244.4); *p* = 0.01], sCD163 decreased by 50.4 ± 145.4 ng/mL [vs. SOC (−9.1 ± 376.3); *p* = 0.02], and BDG decreased by 40.3 ± 115.8 pg/mL [vs. SOC (−15.8 ± 284.2); *p* = 0.03].

In time-varying adjusted models (Table 2), the mean difference in the LC Research Index was 3.2 points lower in the vitamin K2/D3 arm compared to the SOC arm (*p* = 0.04), and female sex had an LC Research Index 2.4 higher (*p* = 0.04) compared to that of male sex. In Table 3, the vitamin K2/D3 arm had nearly two less total number of LC symptoms [−1.7 (−3.2, −0.1); *p* = 0.02] compared to the SOC arm. In Table 3, non-white race had a 2.4 times higher number of LC symptoms compared to white race (*p* = 0.01), and improvement in gut permeability (↓ Zonulin) was associated (*p* = 0.02) with a reduction in the total number of LC symptoms.

## 4. Discussion

For the first time, in this randomized controlled clinical trial in individuals with long COVID, we investigated the effects of vitamins K2 and D3 on long COVID symptomatology, as well as markers of immune activation and gut function. We observed a treatment effect of K2/D3 on reducing the long COVID Research Index and total number of symptoms. These findings highlight the promising impact of vitamin K2/D3 supplementation in reducing long COVID.

Among the most commonly reported reductions in long COVID symptoms, a significant decrease in participants experiencing body pain and post-exertional malaise was observed. This suggests a potential positive impact of vitamin K2/D3 supplementation on muscle-related symptoms. These findings are consistent with previous studies highlighting the possible beneficial effects of vitamin D on muscle health [30], as well as a recent randomized clinical trial demonstrating that vitamin K2 supplementation reduces the frequency, intensity, and duration of nocturnal leg cramps in older adults [31]. Our analysis revealed a significant increase in serum 25(OH)D levels at 24 weeks among individuals in the vitamin K2/D3 group, indicating strong adherence to the study intervention. This is in contrast with the lack of change in vitamin D levels in the control group, highlighting the fact that these participants did not seek supplements outside of study.

To explore the mechanisms behind our findings, it is essential to highlight the key role of inflammation in the pathophysiology of long COVID. In this study, we demonstrated a significant reduction in several inflammatory markers 24 weeks after the intervention in the vitamin K2/D3 group, particularly in the monocyte activation marker sCD163 and the inflammatory biomarker TNF-RI. This aligns with previous research showing that vitamins K2/D3 can modulate the immune system by inhibiting cytokine production and stabilizing immune responses [15,16]. Persistent inflammation is a critical factor in long COVID, making these findings especially relevant. The observed reduction in inflammation suggests that vitamin K2/D3 supplementation may effectively alleviate long COVID symptoms by modulating immune dysregulation. Additionally, prior studies have linked deficiencies in vitamins D and K to more severe acute COVID-19 outcomes [22,23,24,25,26,28], with the severity of acute illness being a known driver of long COVID [32]. Our results confirm the beneficial and safe role of vitamins K2/D3 for long COVID, addressing a crucial therapeutic gap that could be globally implemented.

In addition to improvements in inflammatory markers, we observed a decrease in oxLDL, but not LDL cholesterol, levels over the 24-week intervention period, underscoring the potential positive impact of vitamins K2/D3 on cardiovascular health. This aligns with previous studies outside of COVID-19 by Qasemi et al., which demonstrated the beneficial effects of vitamin D on oxLDL in diabetic patients [33] and the synergistic role of vitamins K2 and D3 in enhancing cardiovascular health [34,35]. Notably, prior research has also identified elevated oxLDL levels in individuals with long COVID, further highlighting the relevance of our findings [9].

Our study also investigated gut permeability and microbial translocation markers to assess potential changes in individuals with long COVID. Previous research suggests that microbial translocation during infection can exacerbate inflammation [11,36]. We previously found, in a separate long COVID cohort, that increased levels of Zonulin, a gut permeability marker, were directly associated with long COVID symptoms [14]. Giron et al. demonstrated elevated levels of beta-D-glucan (BDG) in individuals with long COVID compared to those who never had COVID-19 or those who had recovered without long-term symptoms [13]. Our results aligned with these findings, showing a significant decrease in BDG levels among individuals with long COVID who received the study intervention. This reduction in BDG occurred alongside a decrease in markers of inflammation and, supporting the role of fungal translocation in the ongoing inflammation seen in long COVID. Moreover, the reduction in the total number of long COVID symptoms was associated with a decrease in Zonulin levels, indicating improved gut permeability. These observations support existing hypotheses that gut barrier dysfunction along with viral persistence contributes to long COVID [6,37]. Elevated Zonulin and fungal translocation markers in long COVID patients correlate with systemic inflammation and multi-organ symptoms [13,37]. Thus, improving gut permeability can help restore the intestinal barrier, preventing the translocation of bacterial and fungal products into the bloodstream. This mechanism helps prevent immune activation and systemic inflammation, and promotes a balanced immune response [6,13,14,37]. Our findings further suggest that enhancing gut health could play a key role in reducing long COVID-related inflammation and symptoms.

Furthermore, BMI and fasting lipids remained unchanged from baseline to the end of the study in both groups, suggesting that vitamin K2/D3 supplementation did not significantly affect these metabolic markers. These findings align with a previous meta-analysis that reported no significant changes in BMI or lipid profiles following vitamin K supplementation [38]. In contrast, Radkhah et al. demonstrated, in a meta-analysis, the positive effect of vitamin D supplementation on lipid profile by decreasing triglyceride and total cholesterol and increasing HDL levels [39]. Similarly, Tarkesh et al. found that daily vitamin K2 supplementation (90 µg Menaquinone-7) for 8 weeks significantly decreased serum triglyceride levels, waist circumference, and body fat mass in females with polycystic ovary syndrome [40]. These discrepancies may be due to different intervention durations, dosages, and patient populations. Further research is needed to clarify the effect of vitamin K2/D3 supplementation on metabolic markers.

Regardless of the observed increase in average glucose levels within the vitamin K2/D3 group, our study did not show a significant change in HOMAR-IR among or within each group. These results contradict previous findings showing that vitamin K and D3 supplementation can reduce glucose and HOMA-IR significantly compared to a placebo group [38,39,41].

Despite the encouraging outcomes observed in our study, it is essential to recognize a few limitations that should be considered when interpreting our results. First, our study did not assess the longitudinal change in symptom severity. Future research could benefit from monitoring symptom severity to examine the impact of these vitamins, not only on symptom resolution, but also on the progression of severity. Second, participants randomized to the control arm or SOC arm did not receive a placebo, despite the stability of 25(OH)D overtime in this group compared to the rise in the vit K2/D3 arm, which makes us reasonably certain that participants did not seek these vitamins outside of the study. Third, it is important to acknowledge the potential impact of unmeasured confounding factors, such as diet and physical activity, which were not accounted for in the analysis. Nonetheless, this study provides an important first step to explore the benefit of vitamin K2/D3 supplementation in people with long COVID. Future studies can build upon these findings by including a larger sample size to enhance the robustness and generalizability of the results. Lastly, the intervention in our study was confined to 24 weeks. Longer interventions may be needed to fully alleviate long COVID symptoms and to further investigate the effects of vitamins K2/D3 on maintaining improved outcomes.

## 5. Conclusions

In this clinical trial, we demonstrated that a daily dose of 2000 IU of vitamin D3 and 240 µg of vitamin K2 over 24 weeks could help improve symptoms of long COVID by reducing the total number of symptoms, the RECOVER long COVID Research Index score, and attenuating the hyper-inflammatory state. Vitamin D3 and K2 supplementation was also effective in reducing gut permeability and fungal translocation, both of which are important factors in the pathophysiology of long COVID. Further studies including a larger pool of participants over extended periods are necessary to confirm the efficacy of this intervention for individuals affected by this condition, given its burden on the community and the absence of effective treatments. This safe intervention has the potential for global impact by significantly improving the quality of life and comorbidities for many people dealing with long COVID.

## Figures and Tables

**Figure 1 nutrients-17-00304-f001:**
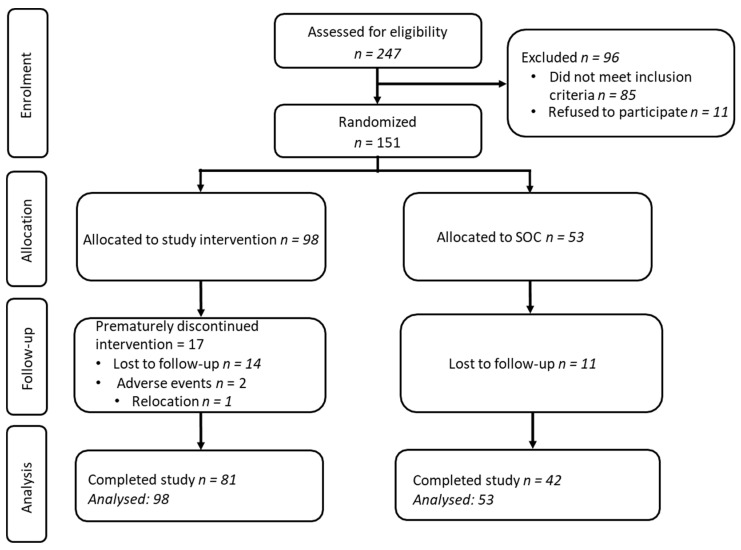
Flow diagram of participant enrollment, allocation, follow-up, and analysis. SOC = standard of care.

**Figure 2 nutrients-17-00304-f002:**
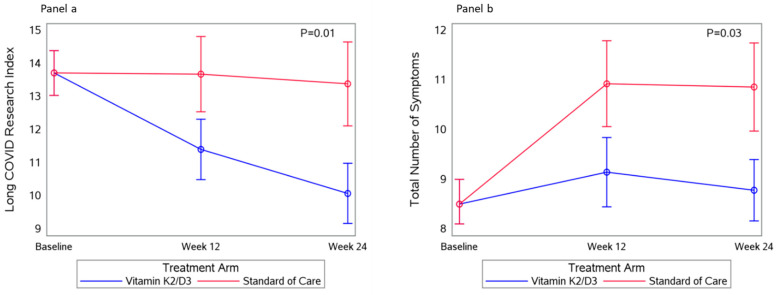
Changes in long COVID Research Index and total number of long COVID symptoms. Panel (**a**): the average change in long COVID Research Index over 24 weeks. Panel (**b**): the average change in the total number of long COVID symptoms over 24 weeks.

**Figure 3 nutrients-17-00304-f003:**
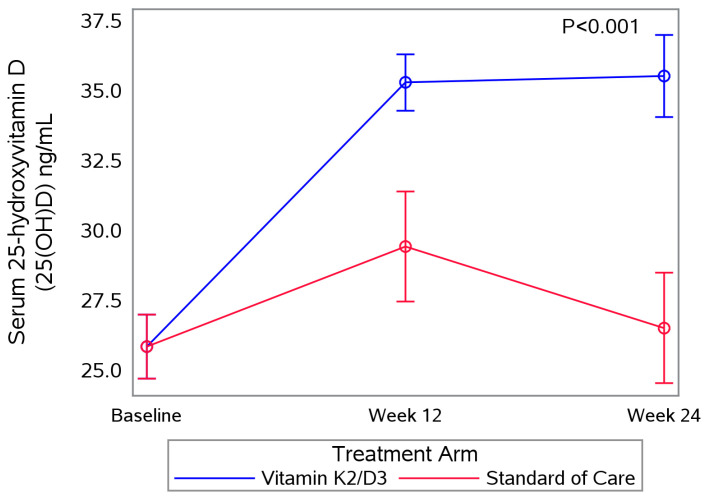
The average change in serum vitamin D between groups over 24 weeks.

**Figure 4 nutrients-17-00304-f004:**
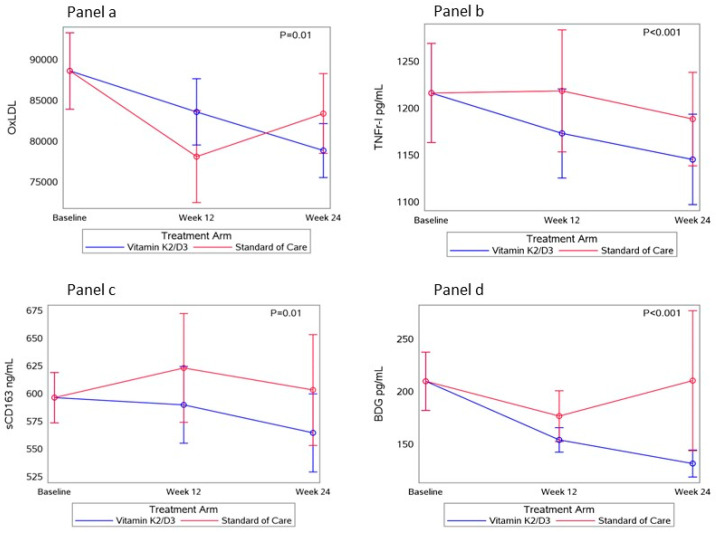
The change in the average levels of OxLDL, TNF-RI, sCD163, and BDG over time between groups. Panel (**a**): changes in the average level of OxLDL between groups over 24 weeks. Panel (**b**): changes in the average level of TNF-RI between groups over 24 weeks. Panel (**c**): changes in the average level of sCD163 between groups over 24 weeks. Panel (**d**): Changes in the average level of BDG between groups over 24 weeks.

**Table 1 nutrients-17-00304-t001:** Baseline characteristics of study participants by treatment allocation.

	K2/D3 (*n* = 98)	Standard of Care (*n* = 53)	*p*-Value
	Median (IQR)/Mean ± std or *n* (%)		
Age (years)	45 ± 12.9	46.2 ± 14.2	0.9
Female Sex	69 (70.4)	38 (71.7)	0.9
Non-white Race *	30 (30.6)	14 (26.4)	0.6
Current Smoker (Yes)	19 (19.4)	9 (17)	0.7
Total Number of Symptoms	8.1 ± 5.5	9.5 ± 6.1	0.1
Long COVID **	42 (42.9)	29 (54.7)	0.2
LC Research Index	11 (6, 17)	16 (8, 22)	0.01
General Phenotype	93 (94.9)	49 (92.5)	0.7
Neurocognitive Phenotype	88 (89.8)	49 (92.5)	0.8
Cardiopulmonary Phenotype	81 (82.7)	49 (92.5)	0.1
Metabolic Markers			
25(OH)D ng/mL	24.8 ± 9.4	27 ± 9.7	0.2
<20 ng/mL	26 (26.5)	16 (30.2)	0.8
≥30 ng/mL	29 (29.6)	13 (24.5)	
BMI (kg/m^2^)	31.2 ± 7.7	31.7 ± 9.3	0.7
Glucose (mg/dL)	102.7 ± 25.6	102.3 ± 26.9	0.9
HOMA-IR	3.8 ± 4.32	4.11 ± 4.29	0.7
LDL (mg/dL)	107.4 ± 33.6	107.8 ± 36.3	0.9
HDL (mg/dL)	57.5 ± 17	55 ± 15.8	0.4
Non-HDL (mg/dL)	131.5 ± 38.8	133.5 ± 42.9	0.7
Triglycerides (mg/dL)	138.2 ± 177.9	128.5 ± 83.2	0.6
Inflammation Markers			
IL-6 (pg/mL)	2.8 ± 3.8	2.5 ± 3.3	0.7
TNF-RI (pg/mL)	1204.5 ± 492.6	1228.7 ± 395.6	0.8
TNF-RII (pg/mL)	2832.1 ± 1485.6	2797.3 ± 1006.1	0.9
hsCRP (ng/mL)	6239.6 ± 10,542	7181.9 ± 9880.9	0.6
I-CAM (ng/mL)	232.6 ± 69.3	244.8 ± 69.6	0.3
IP10 (pg/mL)	153.6 ± 253.2	164.6 ± 348.8	0.8
D-dimer (ng/mL)	485.1 ± 292.1	613.8 ± 454.9	0.07
oxLDL (mU/L)	89,571.8 ± 47,253.9	87,732.6 ± 32,276.4	0.8
sCD14 (ng/mL)	1782.4 ± 512.6	1812.5 ± 389	0.7
sCD163 (ng/mL)	603.4 ± 289.8	583.2 ± 244.3	0.7
Gut Markers			
Zonulin (ng/mL)	1486.9 ± 1616.4	1711.9 ± 1597.1	0.4
BDG (pg/mL)	169.6 ± 156.5	250.5 ± 278.9	0.06
IFABP (pg/mL)	1946.1 ± 1177.3	1992.4 ± 1246.1	0.8
LBP (ng/mL)	16,266.8 ± 8527.7	15,880.5 ± 8341.9	0.8

* Includes African American, Asian, Hispanic, and Other. ** Long COVID = RECOVER definition of having a PASC Research Index value ≥ 12. Abbreviations: LC = long COVID; 25(OH) D = serum 25-hydroxyvitamin D; BMI = body mass index; HOMA-IR = homeostasis model assessment-estimated insulin resistance; LDL = low-density lipoprotein; HDL = high-density lipoprotein; IL-6 = interleukin-6; TNF-RI = tumor necrosis factor receptor-1; TNF-RII = tumor necrosis factor receptor-2; hsCRP = high-sensitivity C-reactive protein; I-CAM = intercellular adhesion molecule-1; IP10 = interferon-gamma-inducible protein of 10 kDa; oxLDL = oxidized LDL; sCD14 = soluble CD14; sCD163 = soluble CD163; BDG = (1,3)-β-d-glucan; IFAB = intestinal fatty acid-binding protein; LBP = lipopolysaccharide-binding protein.

**Table 2 nutrients-17-00304-t002:** Associations with changes in LC Research Index over 24 weeks.

	Unadjusted		Adjusted *	
	Estimate (95% CIs)	*p*-Value	Estimate (95% CIs)	*p*-Value
K2/D3 vs. SOC	−3.1 (−5.3, −0.9)	0.01	−3.2 (−5.5, −0.9)	0.04
25(OH)D ng/mL	0.2 (−1.8, 2.2)	0.8	1.2 (−1.2, 3.6)	0.3
<20 vs. ≥30 ng/mL	−0.5 (−2.8, 1.8)	0.6	1.5 (−2.9, 5.9)	0.7
20–30 vs. ≥30 ng/mL	−0.9 (−2.7, 0.9)		−0.2 (−3.7, 3.3)	
Age at baseline	2.3 (−1.1, 5.7)	0.2	2 (−1.9, 5.9)	0.3
Sex (Female vs. Male)	2.7 (0.05, 4.9)	0.01	2.4 (0.04, 4.7)	0.04
Race (non-white vs. white)	0.2 (−0.1, 0.4)	0.1	2.1 (−0.3, 4.6)	0.08
BMI (kg/m^2^)	−0.002 (−0.1, 0.08)	0.9	3.8 (−0.6, 8.1)	0.09
non-HDL (mg/dL)	−0.5 (−3, 2.1)	0.7	0.9 (−2.6, 4.3)	0.6
I-CAM (ng/mL)	−1.5 (−3.6, 0.6)	0.2	0.1 (−1.6, 1.8)	0.9
D-dimer (ng/mL)	−0.7 (−2, 0.7)	0.3	1.5 (−0.4, 3.5)	0.9
TNF-RI (pg/mL)	−0.7 (−3.4, 2.0)	0.6	−0.7 (−3.1, 1.8)	0.6
OxLDL (mU/L)	−0.5 (−2.6, 1.7)	0.7	−0.8 (−2.5, 1)	0.4
sCD163 (ng/mL)	−0.6 (−2.3, 1.2)	0.5	−0.02 (−1.4, 1.4)	0.9
BDG (pg/mL)	−0.4 (−1.5, 0.7)	0.5	−0.3 (−1.6, 0.9)	0.8

* Adjusted models included treatment arm, age at baseline, sex, race, and changes in BMI, lipids, D-dimer, and BDG. Changes in markers of inflammation (I-CAM, TN-RI, oxLDL, and sCD163) were modeled separately from each other. Abbreviations: SOC = standard of care; 25(OH) D = serum 25-hydroxyvitamin D; BMI = body mass index; HDL = high-density lipoprotein; I-CAM = intercellular adhesion molecule-1; TNF-RI = tumor necrosis factor receptor-1; oxLDL = oxidized LDL; sCD163 = soluble CD163; BDG = (1,3)-β-d-glucan.

**Table 3 nutrients-17-00304-t003:** Associations with changes in total number of LC symptoms over 24 weeks.

	Unadjusted		Adjusted *	
	Estimate (95% CIs)	*p*-Value	Estimate (95% CIs)	*p*-Value
K2/D3 vs. SOC	−1.7 (−3.4, −0.1)	0.03	−1.7 (−3.2, −0.1)	0.02
25(OH)D ng/mL at baseline	−1.2 (−2.5, 0.1)	0.08	1.5 (−0.4, 3.4)	0.05
<20 vs. ≥30 ng/mL	−1.3 (−2.9, 0.2)	0.1	−1.7 (−2.2, −0.03)	0.07
20–30 vs. ≥30 ng/mL	−1.1 (−2.3, 0.1)		−1.3 (−2.6, −0.03)	
Age at baseline	1.9 (−0.5, 4.4)	0.1	1.2 (−1.2, 3.6)	0.3
Sex (Female vs. Male)	1.1 (−0.5, 2.8)	0.2	0.6 (−1, 2.2)	0.5
Race (non-white vs. white)	1.7 (−0.03, 3.3)	0.05	2.4 (0.7, 4.1)	0.01
BMI (kg/m^2^)	−0.4 (−2.4, 1.6)	0.7	1.5 (−0.4, 3.4)	0.1
non-HDL (mg/dL)	0.6 (−1.1, 2.3)	0.5	1.1 (−0.4, 2.6)	0.1
Zonulin (ng/mL) **	−0.8 (−1.6, −0.1)	0.03	−0.7 (−1.4, 0.2)	0.02

* Adjusted models included treatment arm, age at baseline, sex, race, and changes in BMI, lipids, and Zonulin. Changes in markers of inflammation were modeled separately from each other. ** In adjusted models, Zonulin was modeled as a three-way interaction with treatment group and time. Abbreviations: SOC = standard of care; 25(OH) D = serum 25-hydroxyvitamin D; BMI = body mass index. HDL = bigh-density lipoprotein.

## Data Availability

The data supporting the findings of this study are available upon reasonable request from the corresponding author. Clinical Trial Registry Number and Website Where it was Obtained: NCT05356936 https://clinicaltrials.gov/study/NCT05356936?cond=long%20covid&term=NCT05356936&rank=1, accessed on 18 December 2024.

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
