# Peer review of "Vitamins K2 and D3 Improve Long COVID, Fungal Translocation, and Inflammation: Randomized Controlled Trial"

_nutrients, 2025, doi:10.3390/nu17020304_

Round 1

Reviewer 1 Report

Comments and Suggestions for Authors

The manuscript “Vitamins K2 and D3 Improve Long COVID, fungal translocation, and Inflammation: Randomized controlled trial” is very interesting, especially since the pro-inflammatory processes following long-COVID are worrying.

1. Please divide long sentences into shorter sentences, to make it easier to read and understand

2. Some results are repeated both in writing and in the table and figure. They could be simplified to avoid redundancy.

3. In the abstract, include the hypothesis and primary objectives of the study

4. Clearly state the significance threshold (e.g., p < 0.05) in the materials and methods

5. Specify the randomization method used

6. Clarify whether the control group received a placebo or not

7. Explain how adherence to supplements was monitored.

8. Demographic data could have been tabulated with appropriate statistical differences

9. Improve the legend of figures 2, 3, 4, for better understanding. Also add a short explanation

10. As limitations please add the relatively small sample size and potential confounders that were not taken into account (e.g. diet, physical activity)

11. In the discussion please elaborate on the biological mechanisms behind the observed improvements in inflammatory markers and intestinal permeability.

12. Discuss how the findings could influence clinical practice and public health policy, especially given the long-term global burden of COVID.

13. The conclusion should be improved to emphasize clinical relevance and next steps, such as larger studies or broader implementation strategies

Reviewer 2 Report

Comments and Suggestions for Authors

Several years ago, The COVID pandemic explosion forced scientists to analyse the reason for infection, its aggressive propagation, and the possibility of prevention. The last has been extended to new therapeutic strategies and medical force organisation in critical situations. However, the post-COVID patients escape from the mainstream of interests. It should be pointed out that the far effects of the diseases strongly influence the quality of life parameter. In the article entitled: Vitamins K2 and D3 Improve Long COVID, fungal translocation, and Inflammation: Randomised controlled trial have taken into consideration the long COVID influence on gut condition. The gut and microbiome play an important role in the human immune defence system condition as well as in the vitamin level. The population taken into consideration is too small to be significant; therefore, I suggest putting in the title: preliminary studies. Authors should also show the level of Vid D and Vit K at the initial point of their studies (otherwise, the reference point will be missing). The other parameters have been correctly discussed in the manuscript. There is no doubt that the nutrition profile is crucial for gut conditions; therefore, the patient profile should be discussed. The levels Vit K and D present food components should be discussed; it is well known that Vit D has shown antioxidant properties. Moreover, the authors must show the upper safety daily level of Vit D and K intake.

From the technical point (editorial), the article is well written and readable, and the references have been well-selected and cited. The figures need the space decreases.

 In conclusion, the article is interesting, and I believe that the authors will  make an effort to answer my questions.

Reviewer 3 Report

Comments and Suggestions for Authors

The study of the effects of vitamin K2 and D3 supplementation is interesting and potentially useful by demonstrating positive effects of such supplementation on the biochemical markers of long covid.

Was the diet of the participants monitored (or at least the participants were asked about)?

The study was well designed, the criteria of inclusion and exclusion are clearly formulated and the flow of the study is presented. The units of some reported parameters are apparently erroneous and should be corrected.

Detailed remarks:

Dear Authors, please cite literature in brackets in the Introduction (and in the whole text); several times I was mislead thinking that a number belongs to the text (Introduction).

Literature should be formatted according to the MDPI rules

Table 1 and Line 265: What are the units for ox-LDL? Please specify

Table 1 and 3: Zonulin levels: are the units correct? Shouldn’t they be expressed rather in ng/mL? 1700 mg zonulin/mL is physically impossible

The same concerns units of D-dimer concentration (Tables 1 and 3)

Round 2

Reviewer 2 Report

Comments and Suggestions for Authors

The article after corection can be accepted for publication.